# Prognostic Ability of Enhancer RNAs in Metastasis of Non-Small Cell Lung Cancer

**DOI:** 10.3390/molecules27134108

**Published:** 2022-06-26

**Authors:** Jun Liu, Jingyi Jia, Siqiao Wang, Junfang Zhang, Shuyuan Xian, Zixuan Zheng, Lin Deng, Yonghong Feng, Yuan Zhang, Jie Zhang

**Affiliations:** 1Department of Anesthesiology, Shanghai Pulmonary Hospital, Tongji University School of Medicine, Shanghai 200433, China; yirong007@126.com (J.L.); 2031276@tongji.edu.cn (J.J.); 2School of Medicine, Tongji University, Shanghai 200092, China; 1753111@tongji.edu.cn (S.W.); zhangjunfang@tongji.edu.cn (J.Z.); 1833173@tongji.edu.cn (S.X.); 1653144@tongji.edu.cn (Z.Z.); 3Shanghai Key Laboratory of Tuberculosis, Shanghai Pulmonary Hospital, Tongji University School of Medicine, Shanghai 200433, China; 4Shanghai Clinical Research Center for Infectious Diseases (Tuberculosis), Shanghai Pulmonary Hospital, Tongji University School of Medicine, Shanghai 200433, China; 5Normal College, Qingdao University, Qingdao 266071, China; bss613@126.com; 6Department of Pulmonary and Critical Care Medicine, Shanghai Pulmonary Hospital, Tongji University School of Medicine, Shanghai 200433, China

**Keywords:** Non-small cell lung cancer, metastasis, enhancer RNAs, regulatory network, prognosis

## Abstract

(1) Background: Non-small cell lung cancer (NSCLC) is the most common lung cancer. Enhancer RNA (eRNA) has potential utility in the diagnosis, prognosis and treatment of cancer, but the role of eRNAs in NSCLC metastasis is not clear; (2) Methods: Differentially expressed transcription factors (DETFs), enhancer RNAs (DEEs), and target genes (DETGs) between primary NSCLC and metastatic NSCLC were identified. Prognostic DEEs (PDEEs) were screened by Cox regression analyses and a predicting model for metastatic NSCLC was constructed. We identified DEE interactions with DETFs, DETGs, reverse phase protein arrays (RPPA) protein chips, immunocytes, and pathways to construct a regulation network using Pearson correlation. Finally, the mechanisms and clinical significance were explained using multi-dimensional validation unambiguously; (3) Results: A total of 255 DEEs were identified, and 24 PDEEs were selected into the multivariate Cox regression model (AUC = 0.699). Additionally, the NSCLC metastasis-specific regulation network was constructed, and six key PDEEs were defined (ANXA8L1, CASTOR2, CYP4B1, GTF2H2C, PSMF1 and TNS4); (4) Conclusions: This study focused on the exploration of the prognostic value of eRNAs in the metastasis of NSCLC. Finally, six eRNAs were identified as potential markers for the prediction of metastasis of NSCLC.

## 1. Introduction

Lung cancer is the malignant tumor with the highest incidence and mortality rates in recent years. Approximately 1.6 million people die per annum as a result and lung cancer death rates worldwide are estimated to be higher by the World Health Organization (WHO) [1,2]. Non-small cell lung cancer (NSCLC) is the most common lung cancer, consisting of over 80% of cases [3]. NSCLC is mainly treated with surgery, radiotherapy, or chemotherapy [4]. However, only a few patients with early-stage NSCLC can be treated by surgery, which makes the 5-year survival rate for stage IA NSCLC patients reach up to 70% [5]. Patients with more advanced NSCLC are usually treated with chemotherapy or radiotherapy, which means the 5-year survival rate decreases to around 23%. Besides, significant limitations still exist in new treatments, such as immunological and targeted therapies [6]. Therefore, a more comprehensive understanding of the molecular mechanism of progression and metastasis is critical to improve the prognosis of patients with NSCLC.

Enhancers are short genomic regions [7,8] that can modulate gene expression by interacting with promoters [9]. Enhancer RNAs (eRNAs), produced during the transcription process of enhancers, are of functional importance. Their expression levels correlate with enhancer activity [10,11]. In human cells, quite a lot of eRNAs have been found, many of which play critical roles in the meditation of the activation of target genes by transcriptional circuitry [12]. eRNA promotes transcription by regulating the chromatin accessibility near target gene promoters and binding to target gene promoters [13,14], or forming eRNA-proteins complexes that promote enhancer-promoter loops [15]. Many studies reported the role of eRNAs in cancers. For instance, the activation of ESR1 can broadly induce the increase of eRNA transcription in breast cancer [16], indicating that eRNAs are associated with activating oncogenes or oncogenic signaling pathways. Moreover, KLK3e is an androgen-induced eRNA regulating the gene KLK3, it can regulate AR-dependent gene expression in prostate cancer by scaffolding the androgen receptor (AR)-associated protein complex [17], which means that in some cases, tumorigenesis can be promoted directly by oncogene-induced eRNAs. Importantly, tissue and individual patient specificity were found in the expression of eRNAs, indicating eRNAs’ potential clinical utility in diagnosis, prognosis and therapy for cancers [18,19,20].

However, the eRNA regulation mechanisms underlying NSCLC metastasis have not been elucidated, and eRNA targeted anti-cancer agents that can improve the prognosis of NSCLC patients are still insufficient. Here, differentially expressed transcription factors (DETFs), eRNAs (DEEs), and target genes (DETGs) between primary and metastatic NSCLC patients were determined. Prognostic DEEs (PDEEs) were further identified using univariate and multivariate Cox regression analyses, based on which a metastatic NSCLC-specific prognosis prediction model was constructed. Significant immune cells and immune-related pathways were identified using cell type identification by estimating the relative subsets of RNA transcripts (CIBERSORT) [21] and single sample gene set enrichment (ssGSEA) algorithms [22], respectively. Hallmark pathways correlated with key DEEs were quantified using gene set variation analysis (GSVA) [23]. Importantly, six PDEEs (ANXA8L1, CASTOR2, CYP4B1, GTF2H2C, PSMF1 and TNS4), along with DETFs, DETGs, immune cells, immune-related pathways, and hallmark pathways were integrated into co-expression analysis to construct a regulation network. The results above were demonstrated by multi-dimensional validation to explain the mechanisms and clinical significance unambiguously.

## 2. Results

### 2.1. DEG Identification and Functional Enrichment Analysis

The mechanism by which eRNAs promote the transcription of target genes was shown in Appendix A. An analysis process of this study was shown in Appendix A. All clinical baseline information for primary NSCLC samples was summarized in Table 1. The differential expression patterns of 1648 DEGs (506 upregulated DEGs and 1142 downregulated DEGs) between primary NSCLC and metastatic NSCLC samples were illustrated by the heatmap plot (Figure 1A) and volcano plot (Figure 1B). The GO terms, including biological processes (BPs), cellular components (CCs), and molecular functions (MFs) where the DEGs were mostly enriched were the regulation of peptidase activity, extracellular matrix and enzyme inhibitor activity, respectively (Figure 1C). Moreover, amyotrophic lateral sclerosis was the most significant KEGG pathway in which the DEGs were mostly enriched (Figure 1D).

In addition, we can see from the heatmap plot (Figure 2A) and volcano plot (Figure 2B) that 255 DEEs (59 upregulated DEEs and 196 downregulated DEEs) were identified from 5100 eRNAs between primary NSCLC and metastatic NSCLC samples.

### 2.2. Multivariate Prognostic Model Construction and Independent Prognostic Factors Identification

Twenty-four PDEEs were identified by univariate Cox regression analysis (*p* < 0.05) (Figure 2C) and integrated into the multivariate prognostic model. Then, the efficiency of the model was evaluated by an ROC curve (AUC = 0.699) (Figure 3D).

For the evaluation of the independent prognostic value of RS, the formula mentioned in the methods was applied to calculate the RS for each NSCLC patient, and the distribution of NSCLC patients with low and high RS was shown by the risk scatter plot (Figure 3A) and risk line plot (Figure 3B).

Besides, the survival probability of the high-risk group and low-risk group was shown by the Kaplan–Meier survival curve, which indicated a lower survival probability in the high-risk group (*p* < 0.001) (Figure 3C). Furthermore, the distribution of the high-risk group and low-risk group within the principal components (PCs) (PC1, PC2 and PC3) was shown in Figure 3E. Finally, RS was identified as an independent prognostic factor in the univariate Cox regression (hazard ratios (HR) = 76.734, 95% confidence interval (CI) (30.391–193.746), *p* < 0.001) (Figure 3F) and multivariate Cox regression (HR = 1.372, 95% CI (1.271–1.482), *p* < 0.001) models (Figure 3G), which were adjusted by age, gender, and stage. Then, boxplots showed the LncRNA expression of 24 PDEEs in each TNM staging and stage of NSCLC (Figure 4A–D).

### 2.3. Correlation Analysis of PDEEs and Immune Cells

In all samples, the composition of 22 immune cells was estimated by the CIBERSORT algorithm (Figure 4E), Macrophage M2, T cell CD4+ memory resting and Macrophage M1 have a higher proportion. Figure 4F, G illustrated the differential infiltration degree of the immune cells between the primary NSCLC and metastatic NSCLC and co-expression analysis of 22 immune cells, respectively. As is shown in Figure 4F, the infiltration degree of T cell CD4+ naive, T cell CD4+ memory activated, T cell follicular helper, T cell gamma delta, NK cell activated, Macrophage M2, Macrophage M1 and Mast cell resting in metastatic NSCLC is significantly lower than that in primary NSCLC, but the infiltration degree of T cell regulatory (Tregs) and NK cell resting is significantly higher in metastatic NSCLC. Figure 4G showed that T cell CD8+ is positively correlated with NK cell activated and T cell CD4+ memory activated significantly, but Mast cell activated and Mast cell resting, T cell CD4+ memory resting and T cell follicular helper showed significant negative correlations. Furthermore, co-expression analysis between PDEEs and 22 immune cells indicated a significant regulatory relationship among 16 immune cells and 15 PDEEs with a |correlation coefficient| > 0.15 and *p* < 0.05.

### 2.4. Correlation Analysis of PDEEs, DETFs, Immune-Related Gene Sets, and Hallmark Pathways

The expression levels of 30 DETFs were illustrated in a heatmap plot (Figure 5A) and volcano (Figure 5B) plot, 29 immune gene sets were shown in a heatmap plot (Figure 5F). Besides, the heatmap plot (Figure 5C), volcano plot (Figure 5D), together with the bar plot (Figure 5E) showed the differentially expressed gene sets of hallmarks of cancer in GSVA, HALLMARK_PANCREAS_BETA_CELLS and HALLMARK_KRAS_SIGNALING_DN are highly expressed in metastatic NSCLC. Afterward, correlation analysis was performed based on PDEEs with DETFs, immune-related gene sets, and hallmark pathways, respectively.

Finally, several elements aforementioned were considered to have significant correlations which were extracted for the construction of a regulation network. Specifically, this regulation network consisted of pairwise interactions between 24 DETFs and 15 PDEEs with a |correlation coefficient| > 0.20 and *p* < 0.05, pairwise interactions between 24 immune-related gene sets and 13 PDEEs with a |correlation coefficient| > 0.20 and *p* < 0.05, and pairwise interactions between 34 hallmark pathways and 16 PDEEs with a |correlation coefficient| > 0.25 and *p* < 0.05.

### 2.5. Correlation Analysis of PDEEs, DETGs, and RPPA Protein Chips

The heatmap plot (Figure 6A) and volcano (Figure 6B) plot showed significantly differential expression patterns for 17 DETGs. Afterward, correlation analysis indicated strong correlations between 17 DETGs which were selected for the construction of a regulation network based on a |correlation coefficient| > 0.20 and *p*-value < 0.05.

Besides, 38 RPPA protein chips and 11 PDEEs were identified to have significant correlations with the criterion of a |correlation coefficient| > 0.25 and *p*-value < 0.05.

### 2.6. The Construction of NSCLC Metastasis-Specific eRNA Regulation Network

Twenty-eight DEGs were eventually integrated to construct a regulatory network, and the expression levels of them between primary NSCLC samples and metastatic NSCLC samples were shown in the heatmap plot (Figure 7A). Specifically, 6 PDEEs, 17 DETFs, 5 DETGs, 33 RPPA protein chips, 23 hallmark pathways, 13 immune cells, and 21 immune-related gene sets were selected to construct the NSCLC metastasis-specific eRNA regulation network (Figure 7B). Figure 7C showed the results of co-expression analysis among 6 PDEEs, 17 DETFs, 5 DETGs, 33 RPPA protein chips, 23 hallmark pathways, 13 immune cells, and 21 immune-related gene sets. Finally, ANXA8L1 mainly positively regulated TP63 (DETF, r = 0.345, *p* < 0.001), EXT1 (DETG, r = 0.361, *p* < 0.001), PAI1 (RPPA protein chip, r = 0.368, *p* < 0.001), EGFR (RPPA protein chip, r = 0.258, *p* < 0.001), HALLMARK_P53_PATHWAY (hallmark signaling pathway, r = 0.398, *p* < 0.001), and Mast cell resting (immune cell, r = 0.214, *p* < 0.001) in NSCLC metastasis. CASTOR2 mainly positively regulated H2AFX (DETF, r = 0.451, *p* < 0.001), SLC23A2 (DETG, r = 0.316, *p* < 0.001), TFRC (RPPA protein chip, r = 0.318, *p* < 0.001), HALLMARK_G2M_CHECKPOINT (hallmark signaling pathway, r = 0.336, *p* < 0.001) and NK cell resting (immune cell, r = 0.204, *p* < 0.001) in NSCLC metastasis. CYP4B1 mainly positively regulated FOS (DETF, r = 0.302, *p* < 0.001), GTF2IRD2B (DETG, r = 0.250, *p* < 0.001), NAPSINA (RPPA protein chips, r = 0.365, *p* < 0.001), HALLMARK_FATTY_ACID_METABOLISM (hallmark signaling pathway, r = 0.365, *p* < 0.001), Mast cell activated (immune cell, r = 0.359, *p* < 0.001) and Type_II_IFN_Reponse (immune-related gene set, r = 0.269, *p* < 0.001) in NSCLC metastasis. GTF2H2C mainly positively regulated NAIP (DETG, r = 0.469, *p* < 0.001), TTF1 (RPPA protein chip, r = 0.412, *p* < 0.001), HALLMARK_PROTEIN_SECRETION (hallmark signaling pathway, r = 0.323, *p* < 0.001), Macrophage M2 (immune cell, r = 0.179, *p* < 0.001) and iDCs (immune-related gene sets, r = 0.231, *p* < 0.001) in NSCLC metastasis. PSMF1 mainly positively regulated TP63 (DETF, r = 0.320, *p* < 0.001), EXT1 (DETG, r = 0.276, *p* < 0.001), TFRC (RPPA protein chip, r = 0.251, *p* < 0.001), HALLMARK_MYC_TARGETS_V1 (hallmark signaling pathway, r = 0.344, *p* < 0.001) and Mast cell resting (immune cell, r = 0.152, *p* < 0.001) in NSCLC metastasis. TNS4 mainly positively regulated TP63 (DETF, r = 0.289, *p* < 0.001), EXT1 (DETG, r = 0.310, *p* < 0.001), CD49B (RPPA protein chip, r = 0.332, *p* < 0.001), HALLMARK_P53_PATHWAY (hallmark signaling pathway, r = 0.320, *p* < 0.001) and Mast cell resting (immune cell, r = 0.200, *p* < 0.001) in NSCLC metastasis.

Because eRNA-related transcriptional regulatory changes were implicated in the pathological processes of NSCLC, and traditional long-term treatment with drugs may result in refractoriness and unsatisfactory outcomes, it is urgent to find potential inhibitors which target NSCLC-related PDEEs, DETGs, and DETFs. Therefore, the small-molecule bioactive inhibitors for DEGs within the regulatory network in this study were identified based on the CMap database. The heatmap plot (Figure 7D) showed the statistically significant small-molecule bioactive inhibitors in more than 10 types of cancers. The results indicated that irinotecan (enrichment score = 0.996, *p* value < 0.001) may be the best small-molecule bioactive inhibitor that may inhibit NSCLC metastasis by suppressing the expression of key DEGs in this study.

### 2.7. Analysis of Single-Cell RNA-Seq Transcriptomes

Data of the single-cell RNA sequencing (scRNA-seq) from 24 NSCLC samples (GSE153935) were obtained from the GEO database (https://www.ncbi.nlm.nih.gov/geo/query/acc.cgi?acc=GSE153935, accessed on 8 June 2022) for validation of the subcellular locations of six key PDEEs. Fifteen cell clusters and seven cell types (Alveolar, B, NK/T, Endothelial, Epithelial, Fibroblast and Myeloid) were identified by t-distributed Stochastic Neighbor Embedding (t-SNE) analysis (Figure 8A). The heatmap showed the genes that were up- or down-regulated in the 15 clusters (Figure 8B). Figure 8C showed the expression of the major genes of the seven cell types in each cell type. The Cleveland plot (Upper part of Figure 8D) showed the expression of canonical markers in seven cell types and the bar plot (Lower part of Figure 8D) showed the distribution of seven cell types in 24 NSCLC samples, which validated the accuracy of our cell type annotations.

Expressions of five key PDEEs (ANXA8L1, CYP4B1, GTF2H2C, PSMF1 and TNS4) (Figure 9A), three key TFs (TP63, H2AFX and FOS) and four key DETGs (EXT1, GTF2IRD2B, NAIP and SLC23A2) (Appendix A) in seven cell types were displayed in feature plots. The UMAP plot displayed the cell cycle of cells above (Figure 9B), which showed that cells from cluster 1 were mainly in G1 phase and cells from cluster 4 were mainly in S phase. Pairs of ligand and receptor among the clusters above were displayed by the ligand-receptor plot (Figure 9C). All the results above suggested that in NSCLC, ANXA8L1 and CYP4B1 are mainly expressed in Alveolar cells, TNS4 is mainly expressed in epithelial cells, GTF2H2C is mainly expressed in NK/T cells and PSMF1 is significantly expressed in the seven cell types above.

### 2.8. Multidimensional Validation

Multiple databases were utilized for reducing the bias caused by different platforms and to improve the reliability of our results. The key role of PDEEs including ANXA8L1, CASTOR2, CYP4B1, GTF2H2C, PSMF1, and TNS4 in pathogenesis and metastasis of NSCLC were validated based on GEPIA (Appendix A), UCSC xena (Appendix A), The Human Protein Atlas (Appendix A), cBioPortal (Appendix A), UALCAN (Appendix A) and OncoLnc (Appendix A). In GEPIA and UALCAN databases, it was shown that CASTOR2, GTF2H2C, PSMF1 and TNS4 were highly-expressed in Lung Adenocarcinoma (LUAD), whereas ANXABL1 and CYP4B1 were lowly-expressed in the TCGA LUAD cohort (Appendix A); ANXA8L1, CASTOR2, PSMF1 and TNS4 were highly-expressed in the Lung Squamous Cell Carcinoma (LUSC) cohort, while CYP4B1 and GTF2H2C were lowly-expressed in the TCGA LUSC cohort (Appendix A). CYP4B1 (GEPIA, Appendix A; UALCAN, Appendix A; OncoLnc, Appendix A), TNS4 (GEPIA, Appendix A; OncoLnc, Appendix A) were significantly associated with the prognosis of LUAD. CASTOR2 (GEPIA, Appendix A), CYP4B1 (GEPIA, Appendix A; OncoLnc, Appendix A) and PSMF1 (GEPIA, Appendix A; UALCAN, Appendix A) were significantly related to the prognosis of LUSC. Besides, TNS4 (UCSC xena, Appendix A) was significantly associated with the prognosis of pan NSCLC (all *p* < 0.05).

The clinical characteristics, drug responses, and target genes of six key PDEEs were validated by the eRic database (https://hanlab.uth.edu/eRic/, accessed on 6 August 2021) [19]. The results indicated that ANXA8L1 (cholangiocarcinoma), PSMF1 (stomach adenocarcinoma) and TNS4 (head and neck squamous cell carcinoma, lung squamous cell carcinoma, rectum adenocarcinoma, stomach adenocarcinoma) were highly expressed in these cancers. CYP4B1 was lowly expressed in lung adenocarcinoma (Appendix A). ANXA8L1 (kidney renal papillary cell carcinoma) and CYP4B1 (lung adenocarcinoma and bladder urothelial carcinoma) were highly expressed at the early stage of these cancers, but PSMF1 was highly expressed in stage III stomach adenocarcinoma (Appendix A). CYP4B1 was lowly expressed in high-grade bladder urothelial carcinoma (Appendix A). The expression levels of CYP4B1 (lung adenocarcinoma, bladder Urothelial Carcinoma, bladder urothelial carcinoma, and breast invasive carcinoma) showed a significant relationship with different subtypes of these cancers (Appendix A). Furthermore, ANXA8L1 (kidney renal papillary cell carcinoma), CYP4B1 (lung adenocarcinoma) and TNS4 (stomach adenocarcinoma) showed a significant relation with survival in these cancers (Appendix A). Moreover, smoking showed a significant relation with the expression of CYP4B1 in lung adenocarcinoma (Appendix A) (all FDR *p* < 0.05). Appendix A showed the information of target genes and the drug responses of six key PDEEs.

The accessibility in chromatin of key PDEEs was validated by the data from ATAC-seq downloaded from the TCGA database and the results in LUAD and LUSC were shown in Appendix A, respectively. Twenty-four chromosome open regions of genic, intergenic, intron, exon, upstream, downstream and distal intergenic were shown in Appendix A, respectively (open regions were shown as green peaks). The distance between the open regions of all chromosomes and the regions of gene transcription were calculated and shown in Appendix A. Appendix A showed that in the open region, genes were mainly enriched for skeletal system development, centrosome, cell adhesion molecule binding and MAPK signaling pathway in LUAD, and Appendix A showed that the open region genes were mainly enriched for the positive regulation of nervous system development, centrosome, cell adhesion molecule binding and MAPK signaling pathway in LUSC. Then, six key PDEEs were detected to be accessible in chromatin (Appendix A).

The Chip-seq validation was performed by the Cistrome database to determine whether PDEEs are bound to DETFs binding sites, PDEEs and DETFs (ANXA8L1-TP63, CASTOR2-H2AFX, CYP4B1-FOS, GTF2H2C-TP63, PSMF1- TP63 and TNS4-TP63) with the highest correlation were selected for further analysis. Finally, six key PDEEs were determined to be bound to the DETFs binding sites (Appendix A).

## 3. Discussion

NSCLC is the main type of lung cancer [24], whose mortality ranks top 1 all over the world [25]. In general, the prognosis of patients with NSCLC metastasis is not good [6], it is vital to explore the potential biological mechanisms and biomarkers for prognostic and therapeutic targets related to NSCLC metastasis. Multiple eRNAs are implicated in tumorigenesis and metastasis, which were significant therapeutic targets in metastatic NSCLC.

In our study, 24 PDEEs were identified and based on them, the risk score was defined by the multivariate Cox regression model (AUC = 0.699). Furthermore, the risk score was identified as the significant predictive factor for the prognosis of NSCLC patients by Univariate and multivariate Cox regression analysis adjusted by age, gender and stage. In addition, for the exploration of potential mechanisms of NSCLC metastasis, correlation analysis was utilized to construct the NSCLC metastasis-specific regulation network, including key PDEEs, DETGs, DETFs, RPPA protein chips, hallmark signaling pathways, immune-related gene sets, and immune cells. Finally, six key PDEEs including ANXA8L1, CASTOR2, CYP4B1, GTF2H2C, PSMF1 and TNS4 were identified.

ANXA8L1 (annexin A8 like 1) encoded one of the top 10 antigens identified by the majority of serological tests for pemphigus vulgaris patients [26]. High expression of ANXA8L1 was detected to be associated with poor prognosis in cervical squamous cell carcinoma and endocervical adenocarcinoma (CESC) [27], but there were few studies on the relationship between ANXA8L1 and lung cancer.

CASTOR2 (cytosolic arginine sensor for mechanistic target of rapamycin complex 1 (mTORC1) subunit 2) is a subtype of CASTOR, a kind of protein readily detectable in vertebrates [28]. CASTOR2 was found to be related to the regulation of mTORC1 [28,29,30], a central growth controller that integrates diverse environmental inputs to coordinate anabolic and catabolic processes in cells [31], cancer cells might promote growth transformation and tumorigenesis by manipulating CASTOR2, which was reported in Kaposi sarcoma [32].

As a superfamily of enzymes related to phase I drug metabolism, Cytochrome P450 (CYP) was involved in multiple biological processes, such as maintaining calcium homeostasis, fatty acid metabolism, and steroid and cholesterol biosynthesis [33]. CYP4B1 (cytochrome P450 family 4 subfamily B member 1) is an extrahepatic form of cytochrome P450 predominantly and responsible for the bioactivation of multiple protoxins with tissue-specific toxicological effects [34]. CYP4B1 was found to be associated with a variety of cancers [35] and Czerwinski et al. discovered that CYP4B1 from normal and neoplastic lung tissues, compared with normal tissues, had mRNA levels in tumor tissues that were reduced by 2.4 times [36].

GTF2H2C (GTF2H2 family member C) was a subtype of general transcription factor IIH subunit 2-like, related to transcription by RNA polymerase II and DNA nucleotide excision repair [37]. The expression level of transcription factor IIH was identified to be significantly reduced in alveolar macrophages of idiopathic pulmonary fibrosis patients [38]. In fetal lung and placenta, altered methylation may occur in GTF2H2C to repair the DNA damage caused by exposure to smoking [37]. However, the relationship between GTF2H2C and cancer remained unclear.

PSMF1 (proteasome inhibitor subunit 1), the proteasome inhibitor PI31 subunit, was able to bind to the outer rings of the 20S proteasome directly or compete for 20S binding with the activating particles to inhibit the proteasome activities [39,40,41]. High expression of PSMF1 was shown to be associated with better survival of NSCLC patients, hence PSMF1 was considered as an underlying suppresser gene in NSCLC [42], which demonstrated the accuracy and clinical practicality of our hypothesis.

TNS4 (Tensin 4) participated in the cell movement, which was induced by MET and was related to the GPCR signaling pathway. High expression of TNS4 was reported to be associated with poor prognosis in gastric cancer and esophageal squamous cell carcinoma patients [43,44]. Furthermore, the differential expression and abnormal methylation of TNS4 were identified in LUAD patients. It was found that high expression of TNS4 leads to poor prognosis, and TNS4 may be involved in the mechanisms of DNA methylation in LUAD, which means it may be a potential marker for the prognosis of LUAD patients. [45].

Six key PDEEs mainly positively regulated TP63, H2AFX, FOS (TFs), EXT1, SLC23A2, GTF2IRD2B, NAIP (DETGs), PAI1, TFRC, NAPSINA, TTF1, CD49B (RPPA protein chips), P53 pathway, G2M checkpoint, fatty acid metabolism, protein secretion, MYC targets V1 (hallmark signaling pathway), Mast cell resting, NK cell resting, Mast cell activated, Macrophage M2 (immune cell), Type_II_IFN_Reponse, and iDCs (immune-related gene sets).

TP63 (tumor protein p63) was associated with the proliferation, migration, colony formation, and invasion of certain squamous cell carcinomas (SCCs) [46,47]. The expression of H2AFX (H2A histone family, member X) histone was promoted by USP22 (Ubiquitin-specific protease 22), which was involved in the occurrence and progression of LUAD [48]. FOS (Fos proto-oncogene, AP-1 transcription factor subunit) can induce the abnormal proliferation of lung cancer cells [48].

EXT1 (exostosin glycosyltransferase 1) methylation can regulate gene expression and activate the WNT pathway, which affected the proliferation and migration of NSCLC and predicted a poor prognosis [49]. SLC23A2 (solute carrier family 23 member 2) was known to be required for sodium-dependent transporters of vitamin C [50], and several types of cancer were reported to be linked with a deficiency in vitamin C [51]. GTF2IRD2B (GTF2I repeat domain containing 2B) was involved in chromatin structure modification and gene expression regulation [52], mutations in GTF2IRD2B may cause disorders of gene expression regulation and contribute to carcinogenesis [53]. NAIP (NLR family apoptosis inhibitory protein), is a major anti-apoptotic protein and is targeted by miR-1 and miR-145, which induces cell death and contributes to the development of cancer [54].

PAI1 (phosphoribosylanthranilate isomerase 1) promoted glycolytic metabolism [55], and the migration and chemotaxis of cancer cells relies on the energy obtained via enhanced glycolysis primarily [56,57]. EGFR (Epidermal growth factor receptor) plays an important role in the regulation of the proliferation, differentiation, survival and motility of the tumor cells and was found to be highly expressed in over 60% of NSCLCs [58], which indicates that it has the potential to promote NSCLC metastasis. TFRC (transferrin receptor) promoted the proliferation and metastasis of cancer cells by upregulating the expression of AXIN2, which accelerated the development of cancer [59]. NapsinA (napsin A aspartic peptidase) and TTF1 (thyroid transcription factor-1) were reported as the specific clinical diagnosis indexes and prognostic markers for LUAD [60,61,62]. CD49B was shown to be the cell-surface marker for the enrichment of a subpopulation of leiomyoma cells that possess stem/progenitor cell properties [63]. The functions of RPPA and their regulatory relationship with PDEE were shown in Appendix A.

The transcription factor p53 participated in the mechanism of the cell cycle and was reported as an important tumor suppressor [64]. The G2/M checkpoint checked cell size and DNA damage in mitosis of multiple organisms. Disorder of mitotic entry can often cause oncogenesis or cell death [65]. Deregulated anabolism and catabolism of fatty acids metabolic were identified as metabolic regulators that support cancer cell growth [66]. Protein secretion signaling was reported to be associated with multiple biological processes, such as cancer cell migration and invasion [67]. MYC targets v1 signaling and might be related to the higher rates of cell proliferation, resulting in increased aggressiveness of the tumor and worse survival [68]. The tumor microenvironment (TME) was significantly associated with the pathogenesis of lung cancer [69,70]. The proportion of resting NK cells and mast cells resting in lung cancer tumor tissues was reported to be lower than in normal tissues [71]. Mast cells and their activation might result in tumor cytotoxicity and tumor angiogenesis [72]. M2 Macrophages promote the growth, invasion, metastasis, and angiogenesis of cancer cells, and are regarded as one of the main tumor-infiltrating immune cells [73]. IFN-γ (type II IFN) played an important role in anti-tumor responses [74]. IFN-γ receptor impairment or IFN-γ-mediated signal disruption may cause insensitivity to IFN-γ, which may promote the development and progress of tumors [75]. Dendritic cells (DCs) were associated with the regulation of the immune response and played key roles in inducing anti-tumor activity [76], a decrease in immature dendritic cells (iDC) and impaired migration ability may cause cancer [77].

Irinotecan is a topoisomerase inhibitor that causes cytotoxic protein-linked DNA breaks [78], which is a kind of p53-dependent small-molecule bioactive inhibitor [79]. Linked to the results of Cmap, irinotecan may exert an inhibitory effect through the P53 pathway.

Five key eRNAs (ANXA8L1, CYP4B1, GTF2H2C, PSMF1, and TNS4) were found to be expressed in Alveolar, B, NK/T, Endothelial, Epithelial, Fibroblast and Myeloid cells in single cell sequencing analysis and the analysis of the scRNA-seq transcriptomes. The tumor-microenvironment was detected to play an important role in each step of NSCLC metastasis, and the synergistic mechanisms of tumor cells and the microenvironment may provide biomarkers or potential therapeutic targets for cancers [80], which suggests that eRNAs may regulate NSCLC metastasis by interacting with the tumor microenvironment.

The ATAC-seq validation showed that six key PDEEs were chromatin accessible in the NSCLC metastasis state, which indicated that these eRNAs were related to NSCLC metastasis. In addition, through Chip-seq validation analysis, these key PDEEs were found to bind to the binding sites of enhancers and DETFs in this study. Therefore, our study verified that these key eRNAs regulated target genes and mediated NSCLC metastasis by recruiting TFs, which may be implicated in the metastasis of NSCLC.

This is the first study that explored the roles that eRNAs played in NSCLC metastasis, however, several limitations still remained in our study. Firstly, since the samples were all from America, selection bias was inevitable, and the applicability of the prediction model to other countries was uncertain. Secondly, our results were completely based on public databases and have not been verified by our own population studies and experiments. Therefore, multidimensional validation was performed to validate our hypothesis based on various online databases in the multi-omics dimension. Moreover, further cell and animal experimental validation and clinical trials were warranted in future studies, demonstrating correlations between identified key PDEEs and other multi-omics biomarkers, and validating the clinical relevance of our key findings regarding novel eRNA regulatory mechanisms from multiple dimensions.

## 4. Materials and Methods

### 4.1. Data Acquisition

RNA-seq data from 1011 primary NSCLC, clinical information from 829 primary NSCLC, and 258 protein chips were obtained from The Cancer Genome Atlas (TCGA) database (https://tcga-data.nci.nih.gov, accessed on 3 August 2020) [81]. RNA-seq data from 42 metastatic NSCLC were obtained from the MET500 database (https://met500.path.med.umich.edu/, accessed on 4 August 2020) [82], 318 TFs were obtained from the Cistrome database (http://cistrome.org, accessed on 14 August 2020) [83]. Immune gene expression profiles were downloaded from the ImmPort database (https://www.immport.org/, accessed on 19 February 2020). Fifty cancer-related hallmark pathways and 29 immune-related pathways were obtained from the Molecular Signatures Database (MSigDB, Version 7.4) (https://www.gsea-msigdb.org/gsea/msigdb/index.jsp, accessed on 10 September 2020) [84].

### 4.2. The eRNA Expression Data

The list of eRNAs, which was normalized and annotated by Ensemble ID and the corresponding target gene list was available in the eRic (enhancer RNA in cancers) database (https://hanlab.uth.edu/eRic/, accessed on 6 June 2020) [19] in addition to the Chromatin Immunoprecipitation Sequencing (CHIP-seq) results containing acetylated histone H3 lysine 27 (H3K27ac) [85]. In addition, the official gene symbol of each eRNA was identified by the CHIP seeker package based on the location in the hg38 genome [86].

### 4.3. Differential Expression Analysis

Differentially expressed genes (DEGs) between primary NSCLC and metastatic NSCLC were identified by the limma and edgeR algorithm [87,88] with the criteria of a False Discovery Rate (FDR) *p*-value < 0.05 and |log2 Fold Change (FC)| > 1.0; DEEs, DETGs, and DETFs were all identified. To determine the most enriched pathways, the Kyoto Encyclopedia of Genes and Genomes (KEGG) and Gene Oncology (GO) enrichment analysis was applied [89].

### 4.4. Multivariate Risk-Prediction Model Construction and Independent Prognostic Factors Identification

Prognostic DEEs (PDEEs) were screened by univariate Cox regression analysis from identified DEEs. Furthermore, the Least Absolute Shrinkage and Selection Operator (LASSO) regression was used to determine the independent variables with great significance, reducing the over-fitting phenomenon [90]. Afterward, we integrated all PDEEs into a multivariate Cox regression model, and the receiver operator characteristic (ROC) curve was used to evaluate the predictive power of the model. The multivariate Cox regression model formula was applied to calculate the risk score (RS) for each NSCLC sample as follows:RS = β_1_ × NSCLC_1_ + β_2_ × NSCLC_2_ + β_3_ × NSCLC_3_ + …… + β_n_ × NSCLC_n_

In this formula, “n” represented the number of PDEEs in the multivariate model; “β” represented the coefficient of corresponding PDEEs. Then, the median of the RS was used to divide all the NSCLC samples into two groups, including the high-risk group and the low-risk group. Additionally, the survival conditions of the high-risk group and the low-risk group were described by Kaplan–Meier survival analysis. The independent prognostic value of RS was evaluated by univariate and multivariate Cox regression analyses, and the multivariate Cox regression model was adjusted by clinical variables, such as age, gender, and stage.

### 4.5. Identification of PDEE-Related Immune Cells and Immune-Reltaed Gene Sets

The composition of 22 types of immune cells in all samples was evaluated by the CIBERSORT algorithm. To determine the correlations between PDEE signature and immune cell infiltration in NSCLC tissues, PDEE expression matrix data were uploaded to CIBERSORT [21]. Infiltrating immune cells were all extracted for subsequent analysis. Moreover, nonparametric tests were performed to determine associations between the immune cells and different clinical phenotypes.

Additionally, a single-sample gene set enrichment analysis (ssGSEA) was carried out to quantify 29 immune-related gene sets in all samples [22]. Further, Pearson correlation analysis was performed to identify significant correlations between PDEEs and immune cells/immune-related gene sets, where a *p*-value < 0.05 was considered statistically significant.

### 4.6. Identification of Downstream Hallmark Pathways

Gene Set Variation Analysis (GSVA) was utilized to explore potential downstream hallmark pathways of PDEEs. Absolute quantification of 50 hallmark pathways was calculated to determine the differentially expressed hallmark pathways between primary NSCLC samples and metastatic NSCLC samples using the GSVA package [23].

### 4.7. Construction of Metastasis-Specific eRNA Regulation Network for NSCLC

Key PDEEs along with DETGs, DETFs, RPPA proteins chips, 50 hallmark pathways, 29 immune gene sets, 22 immune cells, were integrated into Pearson correlation analysis to construct a NSCLC metastasis-specific eRNA regulation network with the criterion of |correlation coefficient| > 0.20 and *p* value < 0.05 for PDEEs and DETGs; |correlation coefficient| > 0.20 and *p* value < 0.05 for PDEEs and DETFs; |correlation coefficient| > 0.25 and *p* value < 0.05 for PDEEs and RPPA protein chips; |correlation coefficient| > 0.25 and *p* value < 0.05 for PDEEs and hallmark pathways; |correlation coefficient| > 0.20 and *p* value < 0.05 for PDEEs and immune-related gene sets, and |correlation coefficient| > 0.15 and *p* value < 0.05 for PDEEs and immune cells.

Furthermore, the connectivity map (Cmap) database was utilized to identify bioactive small molecule inhibitors with the potential as target drugs for DEGs of the NSCLC metastasis-specific regulation network (https://portals.broadinstitute.org/cmap/, accessed on 25 September 2020) [91,92]. DEGs, including PDEEs, DETGs, and DETFs were used as input data in the Cmap analysis, results of stemness-related DEG analysis in pan-cancer were also integrated into the Cmap analysis [91]. Information on targeting inhibitors could be acquired from the mechanism of actions (MoA) (http://clue.io/, accessed on 25 September 2020) including human cell lines’ transcriptional responses to perturbagens, structural formulas, and protein targets. Therefore, based on MoA, inhibitors that may target NSCLC metastasis-related DEGs in this study were all identified.

### 4.8. Analysis of scRNA-Seq Transcriptomes

scRNA-seq data from 24 NSCLC samples obtained from the GEO database (https://www.ncbi.nlm.nih.gov/geo/query/acc.cgi?acc=GSE153935, accessed on 8 June 2022) were applied to analyze the cellular localization of key PDEEs. The analysis of the integrated data was carried out by the R toolkit Seurat [93]. Genes that were expressed in over 200 single cells and cells with 1500 to 100,000 gene transcripts were selected for further analysis. Moreover, the “vst” method, “FindMarkers” and “FindConservedMarkers” functions were utilized to identify the variable genes, and the marker genes for each cell type were defined. Then, the variable genes above were used to carry out the principal component analysis (PCA) to reduce data dimensionality. The top 15 PCs were selected for further analysis, including Unified Manifold Approximation and Dimensionality Reduction Projection (UMAP) analysis and cluster analysis. The cell clusters, which were based on key PCs, were identified by t-SNE (resolution = 0.50) [94] and absolute values of FDR < 0.05 and log2 (FC) > 0.5 were the criterion for DEG in each cell cluster. Annotation for every cluster was performed by the CellMarker database and the singleR method [95,96]. Visualization of cell cycle stages was performed by the “CellCycleScoring” function and markers of phases. Finally, in different cell types, the pairs of receptor and ligand were identified by the “iTALK” package [97] and visualization of intercellular communication was performed by the “edgebundleR” package (https://github.com/garthtarr/edgebundleR, accessed on 8 June 2022).

### 4.9. Multidimensional Validation

For balancing false positive results and reducing information bias, multi-database external verification was performed based on various online databases including Gene Expression Profiling Interactive Analysis (GEPIA) [98], UCSC xena [99], The Human Protein Atlas [100], cBioPortal [101,102], UALCAN [103] and OncoLnc [104]. In addition, Single Cell Expression Atlas was adopted to identify eRNA expression at the cellular level (https://www.ebi.ac.uk/gxa/sc/experiments, accessed on 30 August 2021) [105]. The assay for transposase-accessible chromatin with high-throughput sequencing (ATAC-seq) data was obtained from the TCGA database to explore the accessibility of eRNAs in chromatin [106]. Chromatin immunoprecipitation sequencing (Chip-seq) data of eRNAs were collected from the Cistrome database to identify specific binding relationships between key PDEEs and DETFs in this study (http://cistrome.org/, accessed on 30 August 2021) [107,108,109,110,111,112,113,114,115,116,117].

### 4.10. Statistics Analysis

All statistical analyses were performed by R version 3.6.1 (Institute for Statistics and Mathematics, Vienna, Austria) (Package: e1071, parallel, preprocessCore, sva, limma, edgeR, ggplot2, survminer, survival, rms, randomForest, pROC, glmnet, pheatmap, timeROC, vioplot, corrplot, ConsensusClusterPlus, forestplot, survivalROC, beeswarm, edgeR, chromVAR, Biostrings, BSgenome.Hsapiens.UCSC.hg38, ChIPseeker, TxDb.Hsapiens.UCSC.hg38.knownGene, clusterProfiler, org.Hs.eg.db, ggplot2, karyoploteR, limma, pheatmap, GSVA, limma, GSEABase, stringr, GEOquery, dplyr, limma, ComplexHeatmap, RColorBrewer, clusterProfiler, tibble, ggplot2, cowplot, ggcorrplot, xlsx, tidyverse, GEOquery, plyr, circlize, ComplexHeatmap, TCGAbiolinks, SummarizedExperiment, dplyr, tidyverse, fgsea, ggplot2, ImmuLncRNA, iTALK, edgebundleR). A two-sided *p*-value < 0.05 was considered statistically significant, and the utilization of the Pearson/Spearman correlation coefficient depended on the results of the normality tests. The machine diagram was drawn on the biorender website (https://biorender.com/, accessed on 2 June 2022).

## 5. Conclusions

This study verified that eRNAs played significant roles in NSCLC metastasis. Moreover, six key PDEEs (ANXA8L1, CASTOR2, CYP4B1, GTF2H2C, PSMF1, and TNS4) were identified as potential markers to predict the prognosis of NSCLC and provide references for the treatment of metastatic NSCLC, which occupied the central position in the NSCLC metastasis-specific regulation network.

## Figures and Tables

**Figure 1 molecules-27-04108-f001:**
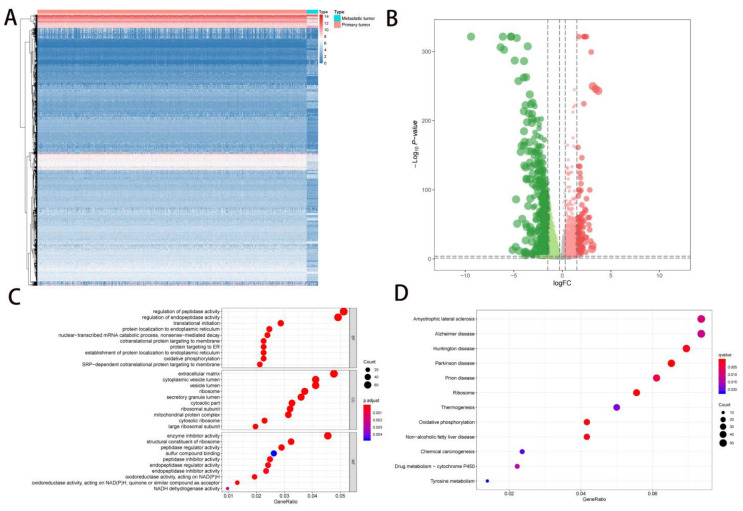
The analysis of DEGs. (**A**,**B**) DEGs were defined between primary NSCLC and metastatic NSCLC. A total of 1648 DEGs (506 upregulated DEGs and 1142 downregulated DEGs) were defined between primary NSCLC and metastatic NSCLC. In the heatmap (**A**), red color represented primary NSCLC and blue color represented metastatic NSCLC. In the volcano plot (**B**), red dot represented upregulated DEGs and green dot represented downregulated DEGs, the two dashed horizontal lines mark the positions of *p*-value = 0.05 and *p*-value = 0.0001, respectively, and the 4 vertical dotted lines mark the positions of log2 foldchange (logFC) = −1.5, logFC = −0.3, logFC = 0.3 and logFC = 1.5, respectively. (**C**, **D**) GO and KEGG pathway enrichment analysis. DEGs enriched in regulation of peptidase activity, extracellular matrix enzyme inhibitor activity and amyotrophic lateral sclerosis significantly. NSCLC, Non-small cell lung cancer; SD, Standard deviation. DEGs, differential expressed genes; NSCLC, Non-small cell lung cancer; GO, Gene Oncology; KEGG, Kyoto Encyclopedia of Genes and Genomes.

**Figure 2 molecules-27-04108-f002:**
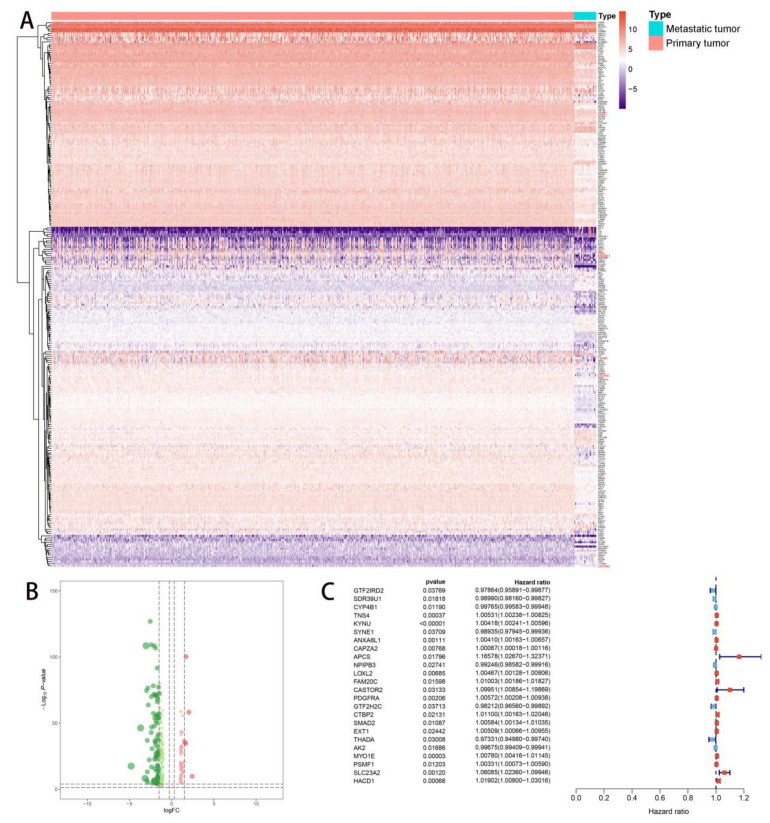
The DEEs identification and DEEs univariate Cox regression analysis. (**A**,**B**) The identification of DEEs. A total of 255 DEEs (59 upregulated DEEs and 196 downregulated DEEs) were identified between primary NSCLC and metastatic NSCLC. In the heatmap (**A**), red color represented primary NSCLC and blue color represented metastatic NSCLC. In the volcano plot (**B**), red dot represented upregulated DEEs and green dot represented downregulated DEEs, the two dashed horizontal lines mark the positions of *p*-value = 0.05 and *p*-value = 0.0001, respectively, and the four vertical dotted lines mark the positions of log2 foldchange (logFC) = −1.5, logFC = −0.3, logFC = 0.3 and logFC = 1.5, respectively. (**C**) DEEs univariate Cox regression analysis. Twenty-four eRNAs were identified as PDEEs (*p* < 0.05). DEEs, differentially expressed eRNAs; NSCLC, Non-small cell lung cancer; PDEEs, prognostic differentially expressed eRNAs; eRNAs, enhancer RNAs.

**Figure 3 molecules-27-04108-f003:**
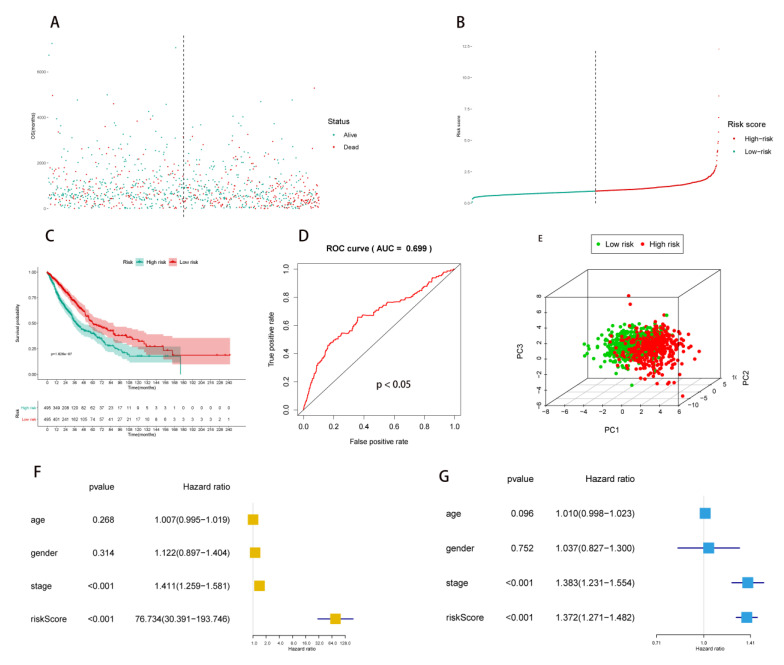
Independent prognosis analysis. RS was calculated by 24 PDEEs integration multivariate Cox model. (**A**,**B**) The scatter plot and line plot illustrated the distribution of RS among all NSCLC patients. (**C**) The ROC curve indicated that the multivariate Cox regression model was of good predictive power (AUC = 0.699). (**D**) The Kaplan–Meier curve showed that RS had prognostic value for NSCLC patients (*p* < 0.05), red line and green line represented low-risk group and high-risk group, respectively. (**E**) The distribution of low-risk and high-risk groups in PC1, PC2 and PC3, red plots represented high-risk groups and green plots represented low risk groups, the proportion of variance for PC1, PC2 and PC3 were 0.141, 0.087 and 0.074, respectively, the cumulative proportion was 0.302. (**F**,**G**) The univariate (HR = 76.734, 95% CI (30.391–193.746), *p* < 0.001) and multivariate (HR = 1.372, 95% CI (1.271–1.482), *p* < 0.001) Cox regression analysis for RS defined RS as an independent prognostic factor for NSCLC patients. The multivariate Cox regression model was corrected by age, gender and stage. RS, risk score; PDEEs, prognostic differentially expressed eRNAs; eRNAs, enhancer RNAs; ROC, receiver operator characteristic; AUC, area under curve; NSCLC, non-small cell lung cancer; HR, hazard ratios.

**Figure 4 molecules-27-04108-f004:**
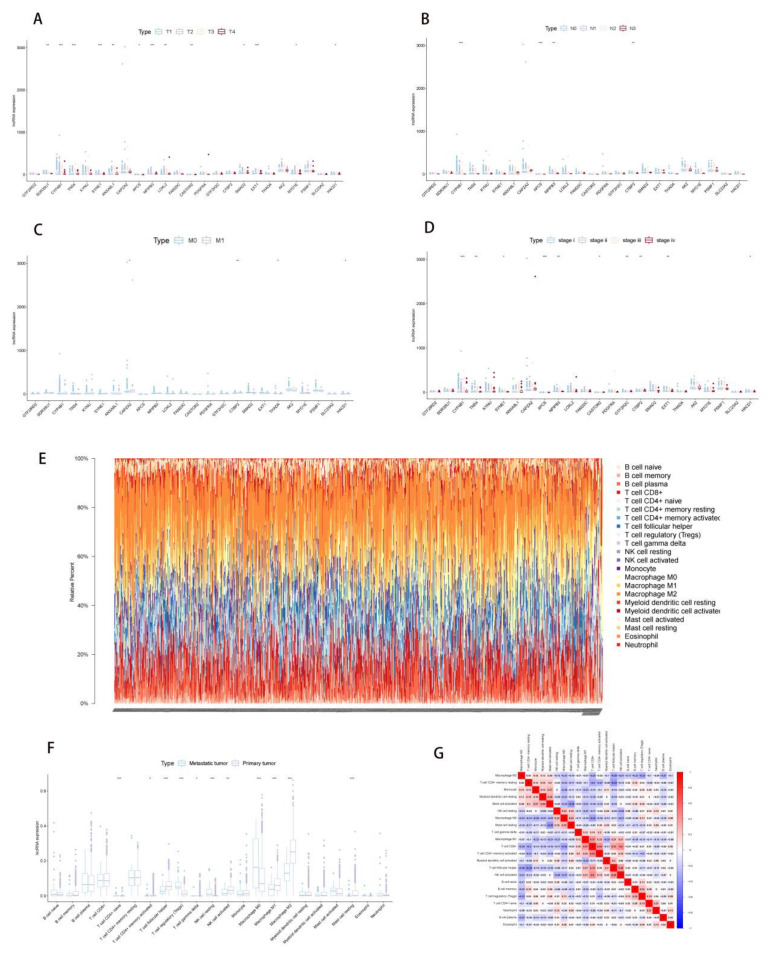
Clinical correlation analysis and immune infiltration analysis. (**A**–**D**) Boxplots showed the LncRNA expression of 24 PDEEs in each TNM staging and stage of NSCLC. (**E**) CIBERSORT algorithm estimated the composition of 22 immune cells in entire NSCLC samples. (**F**) The immune cells that were differentially expressed between primary NSCLC and metastatic NSCLC, purple color and blue color represented primary NSCLC and metastatic NSCLC, respectively. (**G**) Co-expression analysis in entire 22 immune cells. NSCLC, Non-small cell lung cancer.

**Figure 5 molecules-27-04108-f005:**
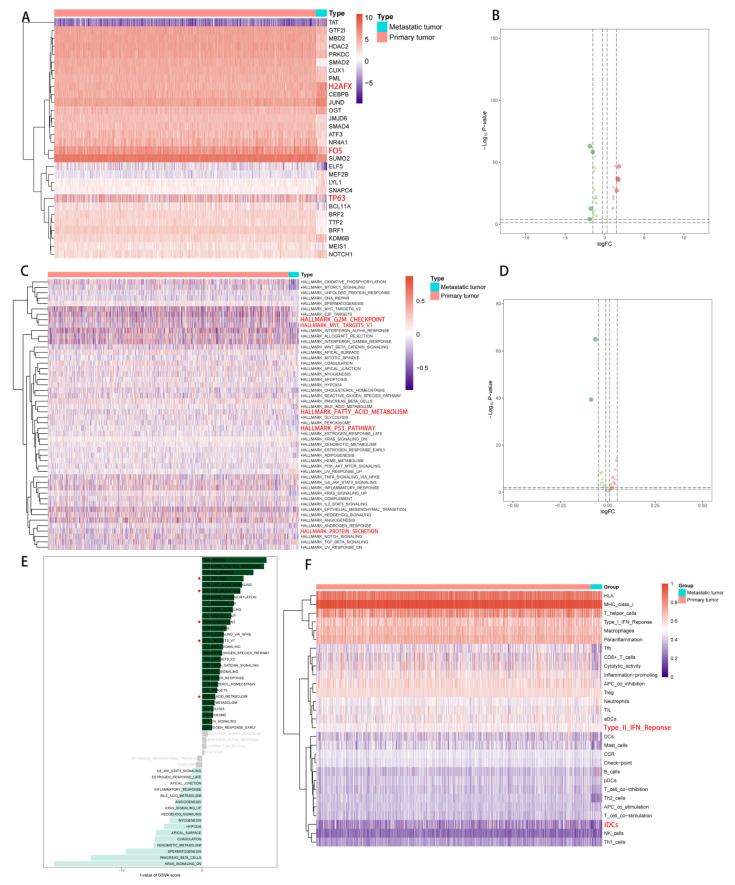
Differential expression analysis among TFs, 50 cancer-related hallmark signaling pathways and 29 immune-related gene sets. (**A**, **B**) Differential expression analysis of TFs. Thirty-one differentially expressed TFs were defined between primary NSCLC and metastatic NSCLC. In the heatmap plot (**A**), red color represented primary NSCLC and blue color represented metastatic NSCLC. In the volcano plot (**B**), red dot represented upregulated TFs and green dot represented downregulated TFs, the two dashed horizontal lines mark the positions of *p* value = 0.05 and *p* value = 0.0001, respectively, and the four vertical dotted lines mark the positions of log2 foldchange (logFC) = −1.5, logFC = −0.3, logFC = 0.3 and logFC = 1.5, respectively. (**C**–**E**) Differential expression analysis of 50 cancer-related hallmark signaling pathways. In the heatmap (**C**), red color represented primary NSCLC and blue color represented metastatic NSCLC. In the volcano plot (**D**), red dot represented the upregulated cancer-related hallmark signaling pathways and green dot represented the downregulated cancer-related hallmark signaling pathways, the two dashed horizontal lines mark the positions of the *p*-value = 0.05 and *p*-value = 0.0001, respectively, and the four vertical dotted lines mark the positions of log2 foldchange (logFC) = −1.5, logFC = −0.3, logFC = 0.3 and logFC = 1.5, respectively. Bar plot (**E**) showed the t-value of the GSVA score among all cancer-related hallmark signaling pathways. (**F**) Differential expression analysis for 29 immune-related gene sets. Red color represented primary NSCLC and blue color represented metastatic NSCLC. TFs, transcription factors; NSCLC, non-small cell lung cancer. 2.6 The Construction of NSCLC metastasis-specific eRNA regulation network.

**Figure 6 molecules-27-04108-f006:**
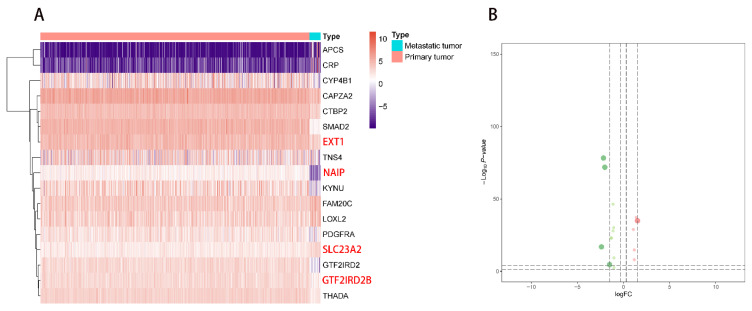
Differential expression analysis of target genes of eRNAs. (**A**,**B**) Differential expression analysis of target genes of eRNAs. Seventeen differentially expressed target genes of eRNAs were defined between primary NSCLC and metastatic NSCLC. In the heatmap plot (**A**), red color represented primary NSCLC and blue color represented metastatic NSCLC. In the volcano plot (**B**), red dot represented the upregulated part and green dot represented the downregulated part in target genes of eRNAs, the two dashed horizontal lines mark the positions of *p*-value = 0.05 and *p*-value = 0.0001, respectively, and the four vertical dotted lines mark the positions of log2 foldchange (logFC) = −1.5, logFC = −0.3, logFC = 0.3 and logFC = 1.5, respectively. eRNAs, enhancer RNAs; NSCLC, non-small cell lung cancer.

**Figure 7 molecules-27-04108-f007:**
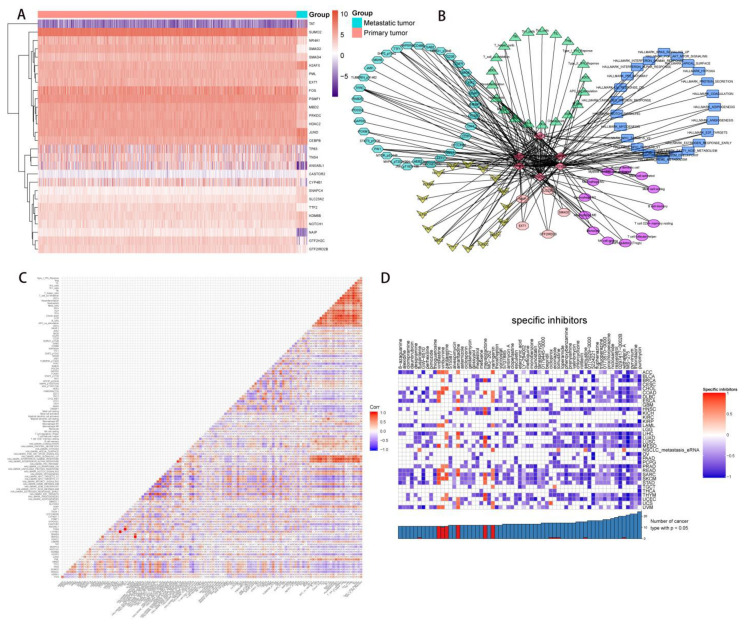
The metastasis-specific regulation network construction of NSCLC and Cmap analysis. (**A**) A total of 28 DEGs were extracted in the construction of the regulatory network. Red color and blue color represented primary and metastatic NSCLC. (**B**) Construction of the NSCLC metastasis-specific regulation network. Six PDEEs, 17 DETFs, 5 DETGs, 33 RPPA protein chips, 23 hallmark signaling pathways, 13 immune cells, and 21 immune-related gene sets in total were selected for the construction of the regulation network. In the network, PDEEs were represented by red rhombus in the center, immune-related gene sets were represented using green triangles, hallmark signaling pathways were represented by dark blue rectangles, immune cells were represented by purple ellipses, DETGs were represented by pink octagons, DETFs were represented by yellow concave quadrilaterals, and RPPA protein chips were represented by light blue hexagons. (**C**) Co-expression analysis among all elements selected in the regulatory network. (**D**) Cmap analysis. Heatmap plot of Cmap analysis showed significant small-molecule bioactive inhibitors in more than 10 types of cancer. Irinotecan (enrichment score = 0.996 *p*-value < 0.001) may be the potential small-molecule bioactive inhibitor for key PDEEs in NSCLC metastasis. DEGs, differentially expressed genes; NSCLC, non-small cell lung cancer; PDEEs, prognostic differentially expressed eRNAs; Cmap, connectivity map; eRNAs, enhancer RNAs.

**Figure 8 molecules-27-04108-f008:**
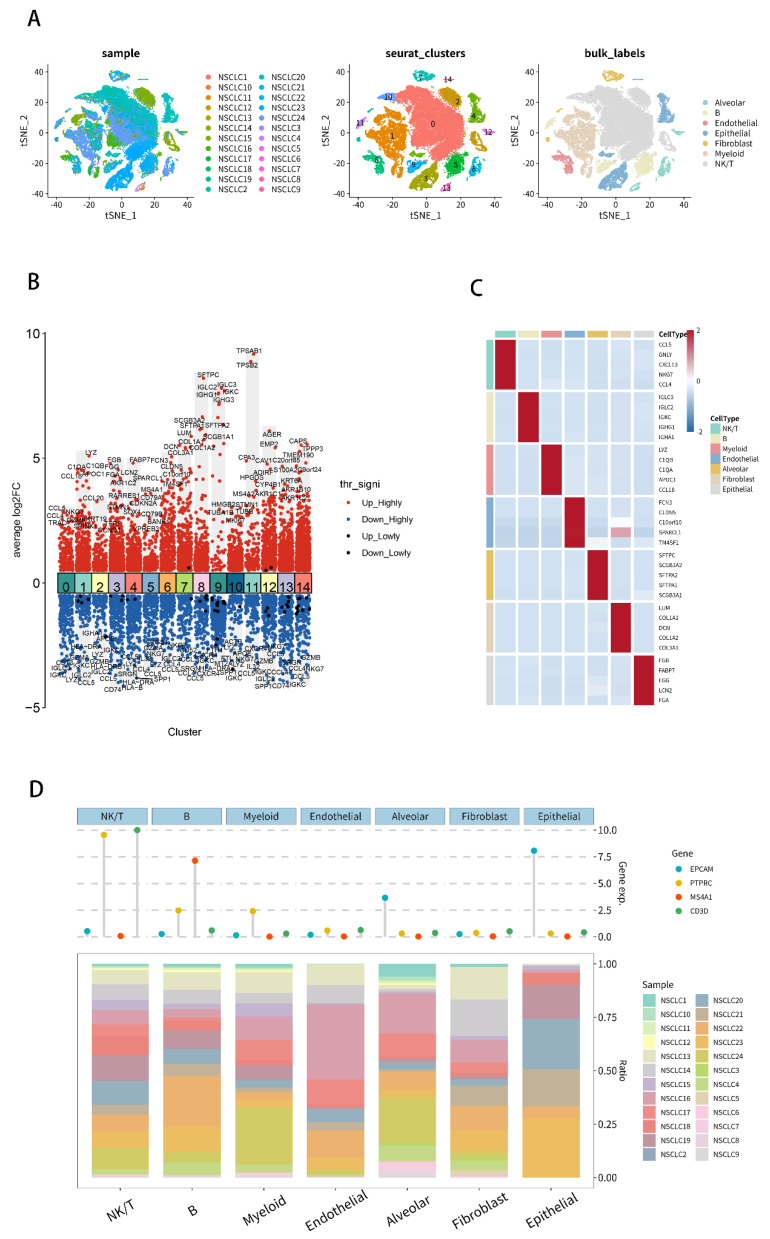
Analysis of scRNA-seq transcriptome for NSCLC to verify the distribution of key biomarkers in various cell types. (**A**) Distribution of 24 NSCLC samples, 15 cell clusters and seven cell types identified by t-SNE analysis. (**B**) Heatmap plot of genes that were up- or down-regulated in the 15 clusters. (**C**) Heatmap plot for the expression of the major genes of the seven cell types in each cell type. (**D**) Cleveland plot (Upper) showed the expression of canonical markers in seven cell types and the bar plot (Lower) showed the distribution of seven cell types in 24 NSCLC samples. scRNA-seq, single-cell RNA sequencing.

**Figure 9 molecules-27-04108-f009:**
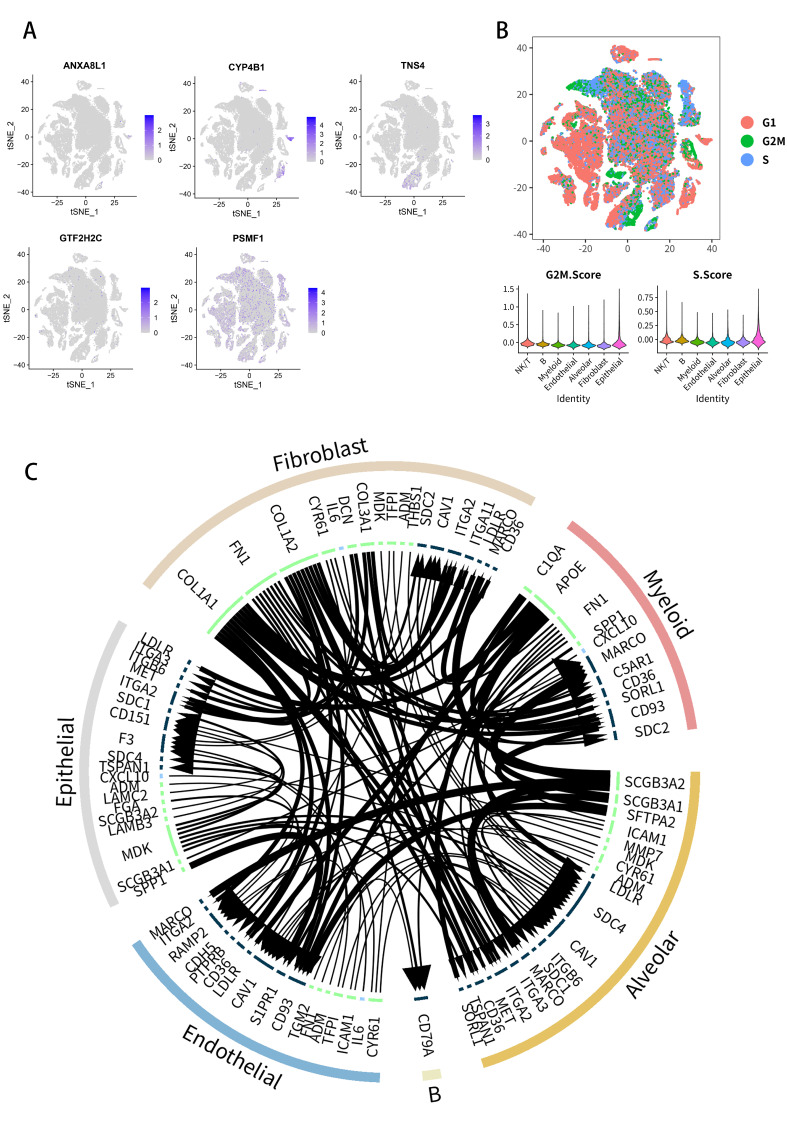
Distribution of five key PDEEs in seven cell types in NSCLC. (**A**) Expressions of five key PDEEs (ANXA8L1, CYP4B1, GTF2H2C, PSMF1, and TNS4) in each cluster. (**B**) Cell cycle of each cell. (**C**) ligand-receptor plot for pairs of ligand and receptor among the seven cell types. NSCLC, Non-small cell lung cancer; PDEEs, prognostic differentially expressed eRNAs.

**Table 1 molecules-27-04108-t001:** Clinical baseline information of 829 primary NSCLC patients.

Characteristics	Total Patients (N = 829)
Age, years	
Mean ± SD	66.30 ± 9.35
Median(Range)	68 (33–87)
Gender	
Female	301 (36.31%)
Male	528 (63.69%)
Stages	
Stage i	415 (50.06%)
Stage ii	242 (29.19%)
Stage iii	144 (17.37%)
Stage iv	28 (3.38%)

## Data Availability

The datasets generated and/or analyzed during the current study are available in the TCGA program (https://portal.gdc.cancer.gov, accessed on 3 August 2020), MET500 database (https://met500.path.med.umich.edu/, accessed on 4 August 2020), Cistrome database (http://cistrome.org, accessed on 30 August 2021), ImmPort database (https://www.immport.org/, accessed on 19 February 2020), Molecular Signatures Database (MSigDB, Version 7.4) (https://www.gsea-msigdb.org/gsea/msigdb/index.jsp, accessed on 10 September 2020) and GEO database (GSE153935) (https://www.ncbi.nlm.nih.gov/geo/query/acc.cgi?acc=GSE153935, accessed on 8 June 2022).

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
