# Peer review of "Prognostic Ability of Enhancer RNAs in Metastasis of Non-Small Cell Lung Cancer"

_molecules, 2022, doi:10.3390/molecules27134108_

Round 1
Reviewer 1 Report
Recent reports are showing that long non-coding RNA including enhancer RNA (eRNA) playing important role in certain cancers. In this aspect this a timely article but presented in an intelligent manner.
Critical Comments:
- Since it is relatively new entity in clinical cancer research, authors may describe properly about the enhancer RNA and how it works as a transcription regulator
- From the data it is not clear enough about the role of eRNA, as it works prognostic or predictive biomarker of standard chemotherapy or targeted therapy or immunotherapy
- Discussion is way too long. Please make it short and describe paragraph by paragraph basis.
- There are some run-on sentences e.g. line #369 to #375
- Need better quality figures and need one line (in the legend) summary of the entire figure for each and every figure
- Is there any comments from authors regarding the expression level of eRNA and co-alterations of driver genes in NSCLC like EGFR, KRAS, KEAP1 etc
- Is there any correlation of eRNA level of expression and the smoking pattern of NSCLC patients
- What is mTORC1 subunit 2(in lane #342)
- Please provide a table of 6 PDEEs and their associated proteins as well as their functions in relation to cancer progression and/or treatment failure
Minor comments:
- Typo error
- Long sentences (it is difficult for the readers)
- Provide full-form of all abbreviations
- If possible make cartoon about enhancer RNA-mediate enhancing transcription and promoter-mediate initiation of transcription
Author Response
Dear editors and reviewers,
Thank you for your comments, please see the attachment to see our response to your comments. Moreover, due to file size limitations and loss of figure quality during the conversion to PDF, please view clear figures in the Word version. We apologize for the inconvenience caused by this issue.
Sincerely yours,
Corresponding author: Yonghong Feng, Yuan Zhang, Jie Zhang
06/14/2022

Reviewer 2 Report
The article discusses a very important topic of searching for prognostic factors of metastasis formation in NSCLC.
After reading the manuscript, I would like to kindly ask you for more details or to revise certain elements of the manuscript.
1. In the manuscript some abbreviations are not explained eg NMSEPs and some abbreviations are not explained when they first appear eg LUAD.
2. The software with which the analyzes were carried out, the results of which were included in the manuscript, were not explained, e.g. ROC curves, survival curves by Kaplan-Meier method, analyses comparing these curves, etc. The authors inform that everything was carried out in the R environment, in such a situation exactly what packages were used in the analyzes should be specified.
3. Table 1: Patient data should be split into primary NSCLC and metastatic NSCLC. What was the exact size of both groups (this information should also be included in the paper)? There is a “$” mark next to the percentage of men - it should be “%”. Below the table are explained abbreviations and between them an OS that is not used in the table.
4. There are a few punctuation errors in the work, eg line 77.
5. In some of the graphs a line should be marked indicating the value of p <0.05 - mainly volcano plots.
6. Lines 109-111: what was the p-value for the ROC curve?
7. Many charts at work are completely unreadable due to too much data or too small font used on them.
8. Lines 118-124: What was the percentage of total variation explained in the PCA and what was the percentage of variation explained by each PC?
Author Response

(The authors gave the same response as above.)

Round 2
Reviewer 1 Report
After revision, it is much more improved MS and ready to be published.